# Imitating Language via
# Scalable Inverse Reinforcement Learning

**Markus Wulfmeier**    **Michael Bloesch**    **Nino Vieillard**    **Arun Ahuja**    **Jörg Bornschein**

**Sandy Huang**    **Artem Sokolov**    **Matt Barnes**    **Guillaume Desjardins**    **Alex Bewley**

**Sarah Maria Elisabeth Bechtle**    **Jost Tobias Springenberg**    **Nikola Momchev**

**Olivier Bachem**    **Matthieu Geist** *    **Martin Riedmiller**

Google DeepMind

## Abstract

The majority of language model training builds on imitation learning. It covers pretraining, supervised fine-tuning, and affects the starting conditions for reinforcement learning from human feedback (RLHF). The simplicity and scalability of maximum likelihood estimation (MLE) for next token prediction led to its role as predominant paradigm. However, the broader field of imitation learning can more effectively utilize the sequential structure underlying autoregressive generation. We focus on investigating the inverse reinforcement learning (IRL) perspective to imitation, extracting rewards and directly optimizing sequences instead of individual token likelihoods and evaluate its benefits for fine-tuning large language models. We provide a new angle, reformulating inverse soft-Q-learning as a temporal difference regularized extension of MLE. This creates a principled connection between MLE and IRL and allows trading off added complexity with increased performance and diversity of generations in the supervised fine-tuning (SFT) setting. We find clear advantages for IRL-based imitation, in particular for retaining diversity while maximizing task performance, rendering IRL a strong alternative on fixed SFT datasets even without online data generation. Our analysis of IRL-extracted reward functions further indicates benefits for more robust reward functions via tighter integration of supervised and preference-based LLM post-training.

## 1   Introduction

In recent years, the imitation of existing human knowledge via large datasets has become a key mechanism underlying increasingly capable and general artificial intelligence systems [17, 45, 9]. Pretraining and supervised fine-tuning phases for large language models (LLMs) predominantly rely on imitation learning, in particular next token prediction via maximum likelihood estimation (MLE). In addition, preference-based fine-tuning is affected by imitation via initial online data generation and optimization objectives such as regularization towards the previously fine-tuned LLM [46, 12].

The field of imitation learning for sequential decision making has a long-standing history for applications such as robotic control [4, 29]. Recently, perspectives to language modeling have shifted towards explicit treatment as a sequential decision making problem – in particular for later stages of model adaptation via reinforcement learning from human feedback (RLHF) [46, 13, 17, 66]. This vantage point opens up new opportunities for the effective use of different data sources and obtaining aligned models that better represent human intent. It includes a broader scope for which data contains information about rewards and preferences as well as the dynamics-aware optimization of each action based on its future impact – all while taking into account computational scalability.

---

*Now at Cohere.

38th Conference on Neural Information Processing Systems (NeurIPS 2024).

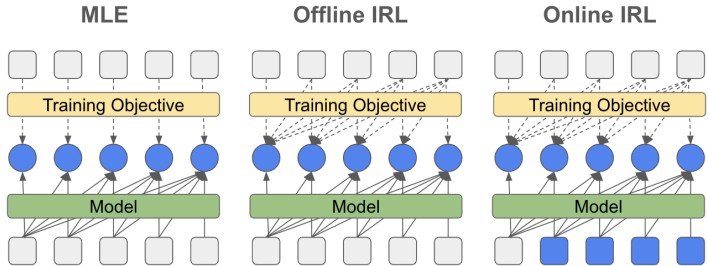

Figure 1: Data usage and optimization flow in MLE, offline and online IRL. Independent of the method, current models use the history of past tokens to predict the next. However, MLE purely optimizes the current output for exact matching the corresponding datapoint while IRL-based methods take into account the impact on future tokens. Online optimization additionally conditions on past model generations rather than the original dataset. Grey and blue objects respectively represent training data and model generations. The impact of future datapoints is often indirect and mediated via learned functions (e.g. the discriminator in GAIL [25] and the Q-function in IQLearn [20]).

Due to the importance of imitation for language modelling, we believe detailed analysis of the underlying problem for sequential decision making is warranted. Despite the widespread perspective of RL for aligning and fine-tuning language models (via RLHF), supervised learning via maximum likelihood estimation for next token prediction remains the dominant component of our imitation learning pipelines due to its simplicity and scalability. However, pure MLE for next token prediction (including "teacher forcing" [64]) can create challenges in autoregressive models, many of which being related to classic challenges with behavior cloning [8], its equivalent in the context of sequential decision making. Compounding errors can occur due to iterative model application creating data sequences which further shift from the training distribution [56, 30], growing increasingly likely for longer sequences [16]. In particular, a model's own samples can cause such distribution shifts and exposure bias [52, 6, 5]. Taking the RL perspective to imitation aims to mitigate these issues via dynamics-aware optimization, where each action is optimized for the impact on the whole future trajectory. It further enables a shift from passive to active learning, where the system actively generates data. Furthermore, best performance of the fine-tuned model is only one metric of importance. Indeed, continued alignment of language models to human preferences requires sampling for a given prompt, collecting human preferences over these completions, and finally aligning the model via preference fine-tuning. Improving the diversity of sampled completions via temperature sampling [7], or using mixtures of past models [61] has been linked to performance improvements. Studying inverse RL, potential divergences, and regularizations provides another angle to increasing diversity [23].

In this paper, we investigate RL-based optimization, in particular the distribution matching perspective to inverse reinforcement learning (IRL), for fine-tuning language models; which can be contrasted with standard MLE as depicted in Figure 1. Our goal is improved understanding of when, and how, IRL can be used as an effective alternative for supervised MLE in the fine-tuning pipeline. The evaluation covers both adversarial and non-adversarial, offline and online methods. We further extend inverse soft Q-learning [20] to explicitly provide a principled connection to classical behavior cloning or MLE. Our experiments range from 250M to 3B parameter models for the encoder-decoder T5 [51] and decoder-only PaLM2 [3] models. Throughout evaluation, we investigate task performance and diversity of model generations illustrating clear benefits of inverse RL over behavior cloning for imitation. A further, principal benefit of RL-centric imitation – close to RLAIF, RL from Artificial Intelligence Feedback – is the natural connection to later RLHF stages via rewards obtained from demonstration data and we take first steps to analyse the value of IRL-obtained reward functions.

Our key contributions are:

- We investigate the RL-centric perspective to imitation for LLMs, extracting rewards and directly optimizing actions for sequence generation instead of individual token likelihood.

- We reformulate inverse soft Q-learning as temporal difference regularized extension of MLE. This explicitly bridges between MLE and algorithms exploiting the sequential nature underlying language generation and enables computationally cheap offline training.

- We compare MLE and IRL formulations including adversarial and non-adversarial, offline and online methods to improve our understanding of imitation in LLMs. Our main results show better or on par task performance, with increased diversity of model generations, in particular demonstrating that key improvements can be obtained via (better scalable) offline IRL.

- Finally, we analyze the extracted reward functions indicating the potential usefulness of IRL to obtain discriminative in addition to generative improvements from demonstrations.

## 2 Methods

Language generation can be modeled as a sequential decision making problem. On a token level, it is the problem of generating the next token $x_i$ given the already generated sequence of tokens $(x_0, \ldots, x_{i-1})$. We thus seek a distribution $\pi(x_i|x_0, \ldots, x_{i-1})$, which we will also refer to as a policy. For a given policy the sequence likelihood $\boldsymbol{x} = (x_0, \ldots, x_N)$ can be computed autoregressively:

$$p(\boldsymbol{x}) = \prod_{i=0}^{N} \pi(x_i|x_0, \ldots, x_{i-1}). \tag{1}$$

The classic maximum likelihood estimation based approach leverages this factorization in order to efficiently train the policy by maximizing the log-likelihood of the training sequences, $\mathcal{D} = \{\boldsymbol{x}^0, \ldots, \boldsymbol{x}^M\}$:

$$\arg\max_{\pi} \sum_{\boldsymbol{x} \in \mathcal{D}} \log p(\boldsymbol{x}) = \arg\max_{\pi} \sum_{\boldsymbol{x} \in \mathcal{D}} \sum_{i=0}^{N} \log \pi(x_i|x_0, \ldots, x_{i-1}). \tag{2}$$

For fine-tuning problems, where a pre-trained model is fine-tuned to a particular set of tasks, the problem can be formulated analogously but may have additional conditioning variables which we will leave out for the sake of clarity.

**Distribution matching.** State-action distribution matching algorithms [25, 20], which are well-established in the field of imitation learning – and can be seen as solving an IRL problem see e.g. [25] – approach the problem in a different manner: They seek to minimize the divergence between the $\gamma$-discounted state-action distribution of the policy $\pi(a|s)$ and the discounted state-action distribution of the expert policy $\pi_E(a|s)$. We define the discounted state distribution for a Markov Decision Process (MDP) as $\rho(s) = (1 - \gamma) \sum_{i=0}^{\infty} \gamma^i P(s_i = s|\pi)$ and the discounted state-action distribution is accordingly defined as $\mu_\pi(s, a) = \pi(a|s)\rho(s)$; where $P(s_i = s|\pi)$ is the probability of seeing state $s$ when acting according to the policy $\pi$. For our autoregressive generation MDP, the state corresponds to the concatenation of already generated tokens (commonly including the prefix or prompt), $s_i = (x_0, \ldots, x_{i-1})$ and the action is the next token, $a_i = x_i$ thus yielding a problem with deterministic dynamics. We omit state and action arguments in the following discussion whenever it is clear from the context.

In order to enable more straightforward algebraic manipulation, the divergence is often combined with a weighted causal entropy term $H(\pi) = -E_{\mu_\pi} \log(\pi(a|s))$. The goal is then to find a policy that minimizes the objective $\mathcal{J}(\pi)$:

$$\mathcal{J}(\pi) := D_f(\mu_\pi||\mu_E) - \lambda H(\pi), \tag{3}$$

where $D_f = E_{\mu_E}[f(\frac{\mu_\pi(a,s)}{\mu_E(a,s)})]$ is an $f$-divergence. Different $f$-divergences have been used in the literature [25, 23]. When taking the example of the reverse KL divergence, with $f(t) = f_{\text{RKL}}(t) = -\log(t)$, we can decompose the objective into a state distribution and an MLE term:

$$\min_{\pi} D_{f_{\text{RKL}}}(\mu_\pi||\mu_E) = \text{KL}(\rho_E||\rho) + E_{\rho_E}[\text{KL}(\pi_E||\pi)]; \tag{4}$$

where KL denotes the KL-divergence which corresponds to maximum likelihood (on the discounted state distribution) for the second part after dropping terms independent of $\pi$. In comparison to MLE, IRL algorithms thus also try to match expert actions but additionally attempt to match the state visitations of the expert. The use of different $f$-divergences can influence mode seeking and mode covering properties of the optimal policy [23].

**Adversarial imitation.** The state-action divergence can be hard to evaluate for two reasons: first it requires many samples from the current policy and second it requires access to the unknown expert densities. In a first step, a variational representation of the $f$-divergence can be leveraged to avoid expert densities:

$$\min_{\pi} \mathcal{J}(\pi) = \min_{\pi} \max_{g:\mathcal{S} \times \mathcal{A} \to dom_{f^*}} -E_{\mu_E}[f^*(g)] + E_{\mu_\pi}[g] + \lambda E_{\mu_\pi}[\log \pi], \tag{5}$$

where $f^*$ is the convex conjugate of $f$ and $dom_{f^*}$ is its domain. Notably, GAIL [25] can be retrieved by using the Jensen-Shannon divergence with its convex conjugate $f^*(g) = -\log(1 - \exp(g(s, a)))$ and defining the discriminator $D(s, a) := \exp(g(s, a))$:

$$\min_{\pi} \max_{D:\mathcal{S}\times\mathcal{A}\rightarrow\mathbb{R}} E_{\mu_E}[\log(1 - D)] + E_{\mu_\pi}[\log(D) + \lambda \log \pi]. \tag{6}$$

**Non-adversarial imitation.** From here, we can re-derive IQLearn [20], but instead consider state value rather than state-action value functions. This representation further allows us to establish a clearer relationship to MLE. Using a change of variable $r : \mathcal{S} \times \mathcal{A} \rightarrow -dom_{f^*}$ with $r(s, a) = -g(s, a)$ and re-arranging the terms, we start by rendering the internal RL-problem more explicit:

$$\min_{\pi} \mathcal{J}(\pi) = -\max_{\pi} \min_{r} E_{\mu_E}[f^*(-r)] + E_{\mu_\pi}[r - \lambda \log \pi], \tag{7}$$

where $r(s, a)$ can be interpreted as an 'implicit' reward function and we omit its arguments in the following for brevity. Due to the convexity of $f^*$ and the concavity of the causal entropy (other terms are linear) this is a saddle point problem [20] and the max-min can be swapped:

$$\min_{\pi} \mathcal{J}(\pi) = -\min_{r} E_{\mu_E}[f^*(-r)] + \max_{\pi} E_{\mu_\pi}[r - \lambda \log \pi], \tag{8}$$

where the second term now corresponds to the discounted cumulative reward of a soft, or entropy regularized, RL problem [74]. The soft-RL problem is well understood [22] and we can use analytical identities regarding its solution for further simplification. In particular, we have that [42]

$$r(s, a) - \lambda \log \pi^r(a|s) = v^r(s) - E_{s'\sim p(\cdot|a,s)}[\lambda v^r(s')] \quad \forall s \in \mathcal{S}, a \in \mathcal{A}, \tag{9}$$

where $\pi^r$ is the optimal policy for the reward $r$ and $v^r$ and $\mu^r$ are respectively the corresponding state value function and discounted state-action distribution[2]. This is applied to the right hand term of Eq. (8) after maximization to obtain (after dropping the arguments for conciseness).

$$\min_{\pi} \mathcal{J}(\pi) = -\min_{r} E_{\mu_E}[f^*(-r)] + E_{\mu^r}[v^r - E_{s'}\gamma v'^r]. \tag{10}$$

Using a telescoping argument [20, 57], we can relate the value of the initial state distribution of a policy to the difference in values on the state-action distribution induced by arbitrary other policies, i.e. $E_{\rho_0}[v^r] = E_{\mu_\pi}[v^r - \lambda v'^r]$. In particular, here we can change the second expectation's state-action distribution to one induced by the expert policy yielding:

$$\min_{\pi} \mathcal{J}(\pi) = -\min_{r} E_{\mu_E}[f^*(-r) + v^r - E_{s'}\gamma v'^r] \tag{11}$$

$$= -\min_{r} E_{\mu_E}[f^*(-r) + r - \lambda \log \pi^r], \tag{12}$$

where we use Equation (9) again in the second line. Here we obtain an explicit MLE term (the last term in the above equation), albeit via the intermediate of the optimal policy of the reward. We currently also have a minimization problem in the reward $r$ and the optimal policy being a function thereof. This can be changed by leveraging the bijective relationship between the reward and the optimal Q-value, $r(q^*) = q^* - E_{s'}[\lambda v'^*]$ [20, 42], or analogously between the reward and a state value and a policy, $r(v^r, \pi^r) = v^r + \lambda \log \pi^r - E_{s'}[\lambda v'^r]$ (using $q^r = v^r + \lambda \log \pi^r$). We can thus reparameterize the problem and optimize the *optimal* state value and policy instead of the reward.

$$\min_{\pi} \mathcal{J}(\pi) = -\min_{v^r, \pi^r} E_{\mu_E}[f^*(-r(v^r, \pi^r)) + r(v^r, \pi^r) - \lambda \log(\pi^r)]. \tag{13}$$

Choosing the $\chi^2$-divergence with convex conjugate $f^*(t) = -\frac{t^2}{4} + t$ is particularly convenient at this point since it can be combined with rescaling the value $v^r_\lambda = v^r/\lambda$ to obtain our reformulated IQLearn objective to be minimized.

$$\mathcal{J}_{\text{IQLearn}}(v^r_\lambda, \pi^r) = E_{\mu_E}[\lambda \underbrace{(v^r_\lambda + \log \pi^r - E_{s'}[\gamma v'^r_\lambda])^2}_{\text{regularization}} \underbrace{- \log(\pi^r)}_{\text{MLE}}]. \tag{14}$$

With this derivation we show the clear relation between MLE and the inverse RL perspectives (expanding on [48]): we can see distribution matching (and thus inverse RL) as performing maximum

---

[2]Note that in practice, we parameterize $v^r_\lambda$ based on the logits of $\pi$ using the identities $\pi^r(a|s) \propto \exp q^r(s, a)$ and $v^r(s) = \log \sum_a \exp q^r(s, a)$ as explained in the appendix.

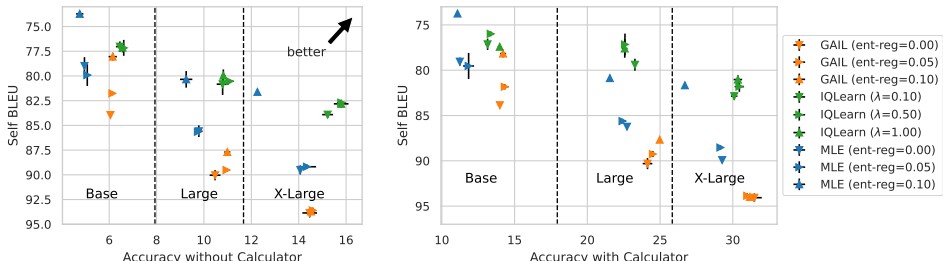

Figure 2: GSM8k results for fine-tuning with MLE, IQLearn, and GAIL across different regularization strengths. In particular MLE shows strong performance reduction with higher entropy cost. Larger models demonstrate higher performance but also stronger self similarity across generations, rendering effective trading of between task performance and diversity highly relevant. Error bars indicate the standard error of the mean after repeating the experiment with 3 different seeds.

likelihood with a dynamics dependent temporal difference regularization term. In contrast to the adversarial setting, this objective does not require samples from the current policy but uses expert samples only[3]. The regularization term couples the learned policy to a value function and favors policies where the log probability of actions matches the difference in state values. An advantage of the above formulation is that it allows annealing of the regularization term where setting $\lambda = 0$ retrieves standard MLE and where by adjusting $\lambda$ the regularization strength can be flexibly increased.

## 3 Experiments

In this section, we evaluate the benefits of different inverse RL based methods in comparison to MLE for training large language models. We assess their impact on task performance, diversity of model generations, and computational requirements. Concretely, we compare MLE-based next token prediction and different IRL methods for fine-tuning LLMs on common benchmarks. In addition, we perform ablations on online[3] vs offline versions of IQLearn, showing results across dataset and model sizes. We finally add analysis of the implicitly learned rewards extracted from SFT data via IRL methods (which bring the potential downstream use to aid RLHF/RLAIF [46, 37] training stages).

These experiments mainly aim to answer the following questions:

- Do IRL methods provide a scalable, effective alternative to MLE for fine-tuning?
- How are different algorithms placed on the Pareto front of task performance and diversity?
- For which task and dataset properties is IRL particularly relevant?
- What is the impact of online data for IRL training?
- How informative are rewards extracted from SFT data?

### 3.1 Algorithms, baselines, and datasets

In addition to naive maximum likelihood estimation for next token prediction, we evaluate the following IRL methods. Generative adversarial imitation learning (GAIL) [25] represents a common adversarial framework training a discriminator to distinguish transitions from agent and dataset and a separate policy. We heuristically adapt the algorithm to mitigate instability for adversarial training [69, 67] including the start for MLE-trained checkpoints and additional loss terms (including MLE) with further details in Appendix A.1.3. IQLearn [20] departs from adversarial learning and our reformulation from Eq. 14 enables us principled control of the temporal difference regularization component to retain stable training. We will use the reformulated offline variant of the algorithm in all experiments and further add an ablation to its online version in Section 3.3.1. Since we compare all methods with respect to task performance and diversity of generations, we additionally evaluate further entropy regularization terms for the MLE baseline; clearly denoted with 'ent-reg' in all plots, corresponding to the respective regularization parameter $\lambda$ in GAIL and MLE (see Appendix A.2 for a more detailed description). In line with previous work on inverse RL for language modelling,

---

[3]An online version of IQLearn is derived in Appendix A.1.2.

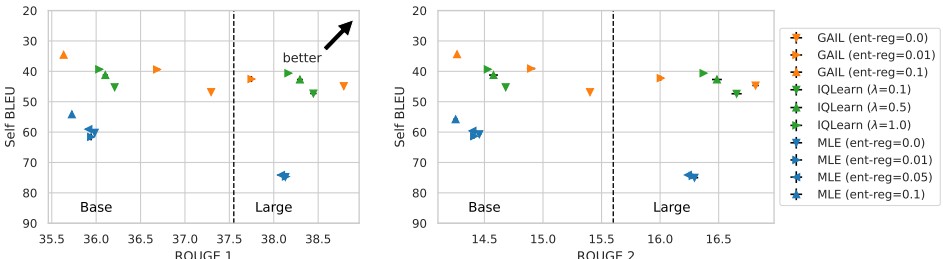

Figure 3: XSUM results for models trained with MLE, IQLearn, and GAIL across different regularization strengths. ROUGE 1 and ROUGE 2 are used as performance metrics on the x-axes with Self-BLEU as diversity measure on the y-axis. Entropy regularizing large MLE and GAIL trained models with 0.1 leads to catastrophic results outside the limits of the plot. Figure 9 in the appendix shows the corresponding plots for ROUGE-LSUM.

we apply a short warm-up phase with pure MLE [15, 65]. Rather than separate experiments or heuristic combinations, the explicit MLE term emerging out of our IRL objective in Section 2 enables principled integration of this mechanism.

We use the following datasets and subsets for ablation in the following sections: XSUM [43], GSM8k [14], TLDR [58], and WMT22 [34]. Unlike parameter-efficient fine-tuning via adapters [27] as used in prior work [15], we focus on the full fine-tuning setting to decouple our analysis from the specifics of adapter-based optimization dynamics [10].

## 3.2 Quality-diversity evaluations

We evaluate both encoder-decoder and decoder-only model classes, respectively using the T5 [51] and PALM2 [3] models. Our main visualizations focus on task performance and diversity of model generations. For task performance, we use the standard metrics for the respective benchmarks (e.g. ROUGE-1 and accuracy percentage). To measure diversity of model generations we calculate self-similarity of generated examples as measured by Self-BLEU [73]. A high score denotes low diversity and vice versa. Using these different axes of evaluation allows us to visualize Pareto fronts between performance and diversity which we will use to assess algorithmic differences. We further add the evaluation via per-token-entropy in Appendix A.3.1.

### 3.2.1 T5 models

We perform experiments with the base, large, and xl T5 models [51] on the XSUM [43] and GSM8k [14] tasks. These models further serve as foundation for our later ablations. We are able to obtain small but notable gains in performance across all tasks, as shown in Figures 2 and 3, in particular math and reasoning tasks show clear improvements in accuracy when fine-tuning with IRL compared to MLE. We hypothesize that specific and shared structure of responses is better exploited via IRL methods. There is a more emphasized boost in the diversity of model generations for IQLearn over MLE. In comparison to prior work [65], we have been able to stabilize GAIL training, with minor changes described in Appendix A.1.3, but are required to start from a checkpoint previously trained via MLE to ensure strong similarity between dataset and model generations starting GAIL training. This only applies to the T5 model class and for PaLM2 models GAIL is highly

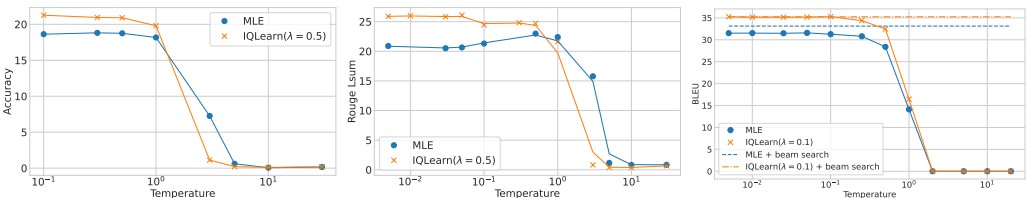

Figure 4: PaLM2 results for various sampling temperatures with MLE and IQLearn. Left: GSM8k, Mid: TLDR, Right: WMT22, including beam search. By propagating sequence information during training, IQLearn reduces inference time dependency on beam search for improving performance.

challenging to stabilize with details in Appendix A.1.3. While entropy terms can further be added to MLE optimization, we hypothesize that better trade-offs between diversity and performance can be obtained via methods able to aggregate information across trajectories to optimize entropy over a different space, complete trajectories rather than just per step policy entropy.

### 3.2.2 PaLM2 models

We also perform experiments with PALM2 models [3], specifically fine-tuning from a pre-trained PALM2 'Gecko' model. We evaluate offline IQLearn on the summarization task TLDR [58], the mathematical reasoning task GSM8k [14], and the large (285M examples) English-to-German translation dataset WMT22 [34]. We limit these experiments to a single choice of the regularization parameter $\lambda$ per task to save computational costs and instead include an analysis of the effect of the sampling temperature parameter during sampling (noting that we similarly observed IQLearn outperforming MLE at varying temperatures for the T5 models). We selected the checkpoints with early stopping for WMT22 as BC was overfitting on the task, unlike TLDR and GSM8K for which we evaluated their latest checkpoints.

Similar to the previous section, we perceive improvements over MLE on all three benchmarks, though for lower accuracy values MLE covers a part of the front. Figure 4 summarizes these improvements, showing the performance of the trained models depending on the temperature used during sampling responses for the test set prompts. These results show a similar behavior between all three tasks, where IQLearn achieves higher performance in a low temperature regime.

## 3.3 Analysis and ablations

We perform additional experiments and post-training analyses to better understand the impact of dataset size, initial checkpoints and computational demands of offline and online algorithms.

### 3.3.1 Computational efficiency and accuracy for online & offline inverse RL

One of the key benefits of (offline) MLE compared with (online) IRL-based methods is its lower computational costs. These are principally related to online rollouts – slow, autoregressive sampling from the model. Our re-formulation of IQLearn results in an algorithm that can be applied offline on a fixed dataset, which underlies all IQLearn results presented, mitigating this limitation. In this section, we additionally present update and sampling times across algorithms[4] and the comparison with the online application of IQLearn (see Appendix A.1.2). Figure 5 demonstrates minimal task performance gains (for the T5 base mode on GSM8k), though considerably improved diversity for model generations. At the same time, Table 1 visualizes the relative cost of sampling in comparison to different algorithm updates, excluding sampling. Note that MLE batch sizes are smaller than IRL ones as we add additional online samples to the batch. Experiment times for online IRL can be reduced, but at the cost of additional hardware, via distributed training with separate sampling infrastructure. The application choice of online or offline IRL finally lies with the practitioner trading of additional computational cost with diversity benefits.

---

[4]Computational resources and time are in practice highly dependent on hardware and implementation but always include crucial, additional sampling costs.

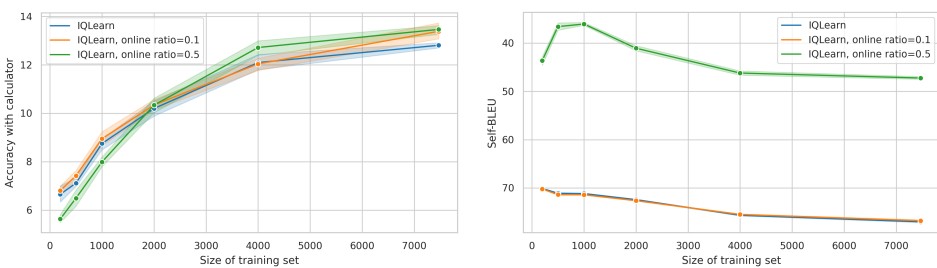

Figure 5: Left: performance of offline and online inverse RL performance with online ratio describing the ratio of offline data used. Right: diversity of model generations. While only showing limited gains in performance, diversity clearly improves.

We provide intuition for low differences between offline and online Inverse RL via toy experiments in Appendix A.3.3. At its core, the dynamics of autoregressive language generation, in particular the concatenation underlying single-turn settings, prevents exact recovery after mistakes, which could be otherwise learned via online IRL. Therefore, extensions of the LLM action space to enable recovery is a fruitful direction for increased benefits from online data with RL or IRL methods [15].

Table 1: Algorithm profiling with computation in milliseconds. 'Sampling' refers to generating a number of sequences equivalent to batch size and often uses equal or more time than updates. These times depend on hardware, implementation and code optimization.

|  | T5-base | T5-large | T5-XL |
| --- | --- | --- | --- |
| MLE Update | $189 \pm 11$ | $422 \pm 17$ | $1031 \pm 22$ |
| GAIL Update | $451 \pm 13$ | $1064 \pm 25$ | $1410 \pm 17$ |
| IQLearn Update | $196 \pm 13$ | $606 \pm 13$ | $1355 \pm 46$ |
| + Sampling (for online IRL) | $443 \pm 45$ | $1345 \pm 211$ | $1823 \pm 327$ |

### 3.3.2 Dataset size ablations

Evaluating training on smaller subsets of GSM8k and XSUM with T5 base is respectively pictured in Figures 6 and 7. Performance improvements are consistent across dataset sizes. The analysis of subsets of XSUM further demonstrates increased robustness against overfitting with IQLearn not showing any of the performance loss over time that plagues MLE in particular with the smallest subsets which cannot be overcome with simple entropy regularization. In line with arguments around the compounding of errors in imitation learning [56], we find that both datasets with longer targets and smaller datasets show stronger task performance gains for IRL.

### 3.4 Reward analysis

In comparison to related work on applying IRL to classical control domains, there is no access to ground truth reward functions underlying the process of data generation. Instead, we measure the correlation between IRL extracted rewards and other task-specific performance metrics. High correlation here tells us how informative a reward function is w.r.t. task performance.

In particular, IQLearn represents learned rewards implicitly via the Q-function as $r_t = Q(s_t, a_t) - \gamma V(s_{t+1})$, and its online version (i.e. with a non-zero mix-in ratio $\alpha$ of on-policy examples, see Appendix A.1.2) additionally exposes the algorithm to (initially, low reward) policy rollouts to help discriminate between them and (high reward) SFT data. In Table 2, we report the Spearman's rank correlation coefficient between accumulated rewards (over complete sampled trajectories for the full validation sets) for online IQLearn ($\alpha = 0.1$) and task-specific metrics. Compared to MLE rewards, which, as expected, are not strongly correlated with any metric, we see a clear increase in correlation pointing to IQLearn incorporating the task-relevant quality into the extracted rewards. We find that using online data is important for consistent correlations across all tasks, in particular as we evaluate over agent generated rollouts. We hypothesize that the comparably lower correlations for GSM8k are likely to be explained by the task's idiosyncratic metric: only the correctness of the final generated numerical expression affects the accuracy calculation, effectively ignoring most of the trajectory (and

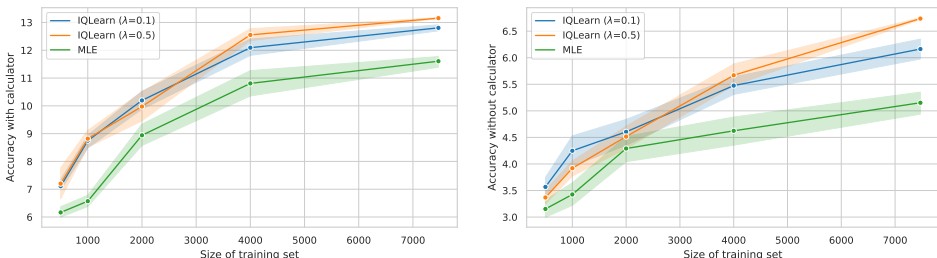

Figure 6: Different subsets of GSM8k. Performance gains persist across dataset scales with larger datasets demonstrating minimal preference for larger regularization coefficients. Each experiment was run with 3 random seeds to obtain uncertainty estimates.

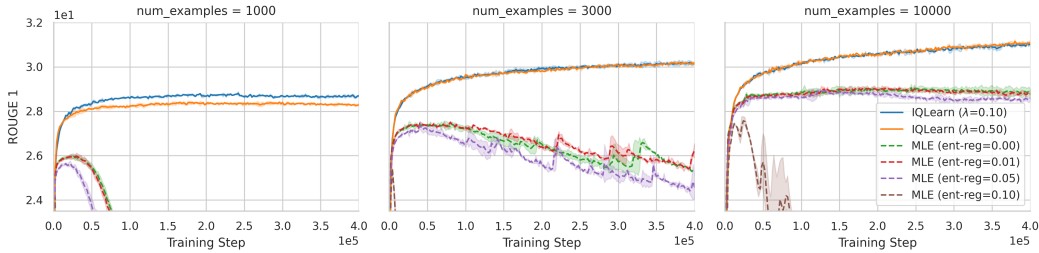

Figure 7: Learning curves for subsets of the XSUM training data. The smallest subsets demonstrate strong overfitting for pure MLE which the TD regularization in IQLearn mitigates. Pure entropy regularization is unable to obtain similar robustness and directly conflicts with task performance.

so its transition's rewards) up to a specific answer segment. This highly targeted reward function becomes harder to learn. Finally, Figure 8 displays how the reward alignment with the task metrics increases with larger TD regularization $\lambda$.

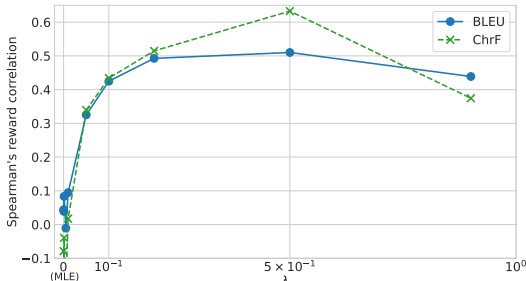

Figure 8: Reward correlation on WMT22 as a function of $\lambda$ for a fixed mix-in $\alpha = 0.1$ for online data.

| Task | Metric | IQLearn | MLE |
|------|--------|---------|-----|
| TLDR | ROUGE-1 | 0.64 | -0.05 |
|      | ROUGE-2 | 0.41 | -0.04 |
|      | ROUGE-Lsum | 0.65 | -0.05 |
| WMT22 | BLEU | 0.43 | -0.05 |
|       | ChrF | 0.43 | -0.01 |
| GSM8k | Acc. w/ calculator | 0.17 | -0.02 |
|       | Acc. w/o calculator | 0.17 | 0.04 |

Table 2: The Spearman's rank correlation for online IQLearn ($\alpha = 0.1$) with $\lambda = 0.1$ (for GSM8k, WMT22) and $\lambda = 0.5$ (for TLDR), compared to MLE (i.e. $\lambda = 0.0$).

## 4 Related Work

**General imitation learning.** Imitation learning assumes a dataset of expert demonstrations, and the aim is to train a policy that matches the expert. There are two broad categories of imitation learning approaches: behavioral cloning (BC) and inverse reinforcement learning (IRL). In BC, a policy is trained using regression to directly mimic the expert demonstrations [49]. This is analogous to supervised fine-tuning of language models via MLE. BC requires sufficient data coverage to perform well, and suffers from compounding errors at evaluation time, as the policy deviates from the state distribution covered by the demonstrations. This can be alleviated by additionally querying the expert for the correct action in the states visited by the agent [56].

**Inverse reinforcement learning.** In contrast, IRL jointly infers the policy and reward function, such that the provided expert demonstrations are optimal under the reward function, and the learned policy maximizes this reward function [44]. By using additional environment interactions beyond the demonstrations, IRL can in theory overcome the compounding errors observed with BC [70]. Note that IRL is related to RL from human feedback (RLHF) but differs in key aspects: IRL also learns both a reward and policy, but extracts information from demonstration and agent data rather than paired preference data [31]. The game-theoretic approach to IRL treats the optimization problem as a zero-sum two-player game [60]. A subset of recent IRL methods can be seen as combining game-theoretic IRL with entropy regularization of the policy, where the doubly-nested optimization is implemented with a classifier (GAIL [25], DAC [35], AIRL [19]), implicit reward functions (ValueDICE [36], IQLearn [20]), or a Lagrangian dual objective (PPIL [62]). The stable, large-scale application of inverse RL methods has been a persistent goal throughout these developments. The classical requirement for complete RL optimization before updating the reward function has presented a limitation [75] which can be overcome via abstracted [68, 9] or linearized models [18], iterative adversarial training [25, 19] or lastly saddlepoint-based value function based formulations [20]. We expand on the insights of the latter to evaluate the competitive performance of computationally cheap offline IRL and emphasise the connection between MLE and IRL [48].

**Imitation learning for language modeling.** Understanding language modeling as an imitation problem has been previously explored. Indeed, the link can already be made from MLE, commonly referred to as Behavioral Cloning (BC) [8] from an imitation perspective. Although the link to imitation has been made explicit recently, either theoretically [59], or in the case of distillation and divergence minimisation [1, 28, 40, 26, 39], some works had already tackled the issues of MLE with imitation-like techniques. For example, adversarial training of text generation an alternative to MLE was first proposed in SeqGAN [71], and followed-up by a series of work using GANs for text generation [32]. These methods have been shown to work only in the temperature 1 regime [11], a possible shortcoming that we address in Section 3. Then, leveraging the literature of imitation learning, GAIL was successfully adapted to language [65], showing an improvement over MLE. Closer to our contributions, IQLearn is utilized for language in SequenceMatch [15]. Key differences to our work include the reformulation as temporal difference regularized MLE, large-scale analysis including other inverse RL methods and focus on computational costs via the application of offline IQLearn. Indeed, SequenceMatch requires the use of online data, via the introduction of the "backward" token, that allows the model to change a previously chosen token during sampling.

## 5 Discussions

Our investigation focuses on diversity measures such as Self-BLEU or model entropy which are easily calculable but limited with respect to their ability to describe the impact on later training stages. Future evaluation and practical application will demonstrate if the increased diversity is relevant to RLHF such as for human raters in preference data evaluation or improved exploration during subsequent RL optimization [53].

The field of imitation learning has led to a gamut of algorithms, many of which are intuitively simple to implement with existing RL or RLHF infrastructure (e.g. [54, 63, 24, 50]). Ease of adaptation and hyperparameter tuning have principal impact on our practical algorithm choices and the methods and extensions discussed in this work enabled quick first results and iteration. Looking forward, the evaluation and adaptation of further imitation learning methods to LLMs is likely to lead to fruitful results in the coming years. While our analysis focuses on specific algorithms, our key arguments should be seen in the light of the benefits of underlying mechanisms rather than the specific methods.

The sampling-free application of RL mechanism can eventually extend to even larger datasets such as pretraining data, domains with high requirements for computational efficiency. While these datasets are, in a certain sense, less optimal and representative of preferred model behavior, different aspects of our analysis have the potential to generalize, such as increased efficiency for modelling sequential decision making or increased diversity of model generations. Pretraining and other sub-optimal data could further be used as additional data source for IQLearn and related algorithms [41, 33, 57] instead of autoregressive sampling data.

Finally, RLHF's key role lies in the alignment of models with respect to user preferences. By integrating SFT data into RL-based optimization, we hope to expand data used for preference description from paired comparisons to individual demonstrations. Integrating generative and discriminative information from different data sources into a unified RL-based framework for unified LLM post-training has further practical potential as previously demonstrated for the generative side [46] with first steps towards reward transfer [38]. Our reward analysis in Section 3 provides an initial signal about IRL-extracted reward functions including crucial information about LLM task performance.

## 6 Conclusions

This paper presents a detailed investigation of the potential of IRL algorithms for imitation in language model tuning focusing on performance, diversity, and computational requirements. We introduce a reformulation of IQLearn which enables principled interpolation between robust, standard supervised fine-tuning and more effective IRL algorithms. Our experiments demonstrate particularly strong gains for IRL on the Pareto front of task performance and diversity of model generations. While prior work primarily focused on online IRL, we demonstrate that computationally cheaper offline IRL, without the requirement of online sampling, already obtains crucial performance gains over MLE-based optimization. Additional correlation analysis between IRL-extracted rewards and performance metrics further emphasises the potential to obtain more accurate and robust reward function for language modelling. We hope this work will help to pave the way for better compromises between data and compute efficiency via RL-based algorithms across the complete LLM training pipeline.

## Acknowledgments and Disclosure of Funding

We extend our gratitude to Daniele Calandriello, Doina Precup, and our anonymous reviewers for their invaluable feedback on the final manuscript drafts. We also would like to acknowledge the contributions of our internal engineering teams, whose robust and flexible RLHF infrastructure significantly accelerated algorithmic development and evaluations. Special thanks are due to Piotr Stanczyk and Leonard Hussenot for their swift and effective solutions to numerous infrastructure challenges. Finally, we deeply appreciate the strategic feedback and guidance provided by Satinder Singh, Dale Schuurmans, and David Silver.

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

# A   Appendix

## A.1   Implementation Details

### A.1.1   IQLearn

For IQLearn, fine-tuning starts with a policy which is subsequently finetuned as a Q-function. In order to do so, we take the logits underlying the LLM softmax layer and continue training as Q-values. In entropy regularized RL, we can build on the equality between optimal policy $\pi^*$ and optimal Q-function $Q^*$ via $\pi^*(a|s) = 1/Z_s \exp Q^*(s, a)$ with normalization factor $Z_s = \sum_{a' \in A} \exp Q^*(s, a')$. We further obtain $V$ via $V(s) = \log \sum_{a \in A} \exp Q(s, a)$.

Due to the translation invariance of the softmax function, the there can be a considerable state-based offset between initial logits and final values. In other words, with $Q(s, a) = V(s) + A(s, a)$ only the advantage function $A$ has to be accurately reflected by the initial policy logits, while the state-based value $V$ can be arbitrarily inaccurate. In practice, we can add further KL regularization, or separate the representation of state and advantage function, to stabilize training during the additional identification of correct offsets but found it to be unnecessary for our experiments in comparison to recent work [15].

We handle terminal states by setting their values to zero in line with the original IQLearn paper [20]. Improvements such as learned values for the terminal states [2, 15] did not contribute to improved performance in this setting.

Table 3: IQLearn and MLE hyperparameters

| Hyperparameter | Value |
|---|---|
| Learning rate T5 | 1e-4 |
| Learning rate PaLM2 | 1e-4 |
| Warmup steps | 2000 |
| Batch size T5 (base/large/xl) | 32/32/16 |
| Batch size PaLM2 | 16 |
| Random seeds / experiment | 3 |

### A.1.2   Online IQLearn

We also implemented an online version of IQLearn, which makes use of additional (non-expert) samples. In contrast to IQLearn [21] which makes use of these examples to estimate the initial value, $E_{\rho_0} v(s)$ (which is equivalent between online and offline data when the prompt is the same as in our case), we integrate the additional samples to relax the distribution matching loss:

$$\min_\pi D_f((1 - \alpha)\mu + \alpha\mu_B || (1 - \alpha)\mu_E + \alpha\mu_B) - \lambda H(\pi), \tag{15}$$

where $\mu_B$ is the discounted state-action distribution of the additional samples and where $\alpha \in [0, 1)$ is the strength of the mix-in. The problem is thus relaxed and allows the policy to match the mix-in distribution $\mu_B$ in case the expert is too difficult to match.

From the adapted distribution matching loss the same steps can then be taken as in Section 2. This results in the generalised loss:

$$-\min_{v^r, \pi^r} E_{(1-\alpha)\mu_E + \alpha\mu_B}[f^*(-r(v^r, \pi^r)) + r(v^r, \pi^r)] - E_{\mu_E} \lambda \log(\pi^r). \tag{16}$$

with

$$r(v^r, \pi^r) = v^r + \frac{\lambda}{1 - \alpha} \log \pi^r - E_{s'} v'^r \tag{17}$$

This formulation strongly relates to the offline version of the algorithm but additionally applies the temporal difference based regularization term on the additional non-expert transitions. A similar algorithm is derived in Appendix D.1 [57] with strong results on classical, continuous control domains of its state-action value function based version. It further connects to prior methods enabling the integration of sub-optimal data into imitation learning [41, 33] with a useful overview on the broader field of related methods in [57].

### A.1.3 GAIL

For GAIL, both the policy and discriminator are represented via separate networks initialized from the initial, pre-trained, LLM. Policy optimization is performed similar to PPO with an A2C update, with re-scaled advantage and KL constraint to the initial policy. The KL constraint has a weight hyperparameter that is annealed over 10,000 steps to a final cost weight displayed below. The value network is also initialized from the initial LLM. The discriminator is trained with a cross-entropy objective. The reward is re-shaped from the the discriminator output to be a positive reward: $r_t = \log(1 + \exp(D(s_t)))$.

We explore the heuristic combination of GAIL with standard MLE training by using a weighted combination of GAIL and MLE losses for the policy. While this is not required for GSM8k, it leads to considerable improvements for XSUM.

Policy, value function and discriminator are all updated with the Adam optimizer, with a constant learning rate and linear warm-up. The discriminator is updated after every step of policy optimization.

Table 4: GAIL hyperparameters

| Method | T5-base | T5-large | T5-xl |
|---|---|---|---|
| Batch size | 32 | 32 | 16 |
| Learning rate | 1e-4 | 1e-4 | 1e-4 |
| Warmup steps | 2000 | 2000 | 2000 |
| KL strength | 1e-3 | 1e-3 | 1e-3 |
| Random seeds / experiment | 3 | 3 | 3 |

## A.2 MLE & Entropy Regularization

As mentioned in the experiments section, we compare our methods to an entropy regularized version of MLE, to disentangle between the imitation contibution and the regularization. This algorithm simply follows the objective

$$\min_\pi E_{\mu_E}[-\log(\pi) - \lambda \mathcal{H}(\pi)], \tag{18}$$

where we compute the entropy at each token of the sequences form $\mu_E$.

### A.2.1 Computational Requirements

Our experiments with T5 models use TPU v3 infrastructure and are running between approximately 3 days and 2 weeks. Our experiments with PaLM2 models use TPU v4 infrastructure and are running under 1 week.

## A.3 Additional Experiments

We include a set of further experiments to complement the results in the main paper.

### A.3.1 Quality-Diversity Evaluation

In addition to the Self-BLEU metric, we further add plots for model entropy and add the ROUGE-LSUM results in Figure 9. Further performance metrics like MAUVE [47] and BertScore [72] can be of use in the future to further represent human judgement and preferences.

### A.3.2 PALM2 Additional Results

We complement results on the temperature sweep over PALM2 models in Figure 10, with additional metrics for GSM8k and TLDR (accuracy with calculator and rouge scores). This confirms the results of the main paper experiment, showing that IQLearn can consistently outperform MLE in accuracy, even when sampling with a lower temperature than the training one.

### A.3.3 Toy Experiments for Offline and Online IRL with Autoregressive Generation

The toy scenario displayed by the MDP in Figure 11 is used to represent a key difference between many classical control settings and autoregressive language generation. The agent starts in the left

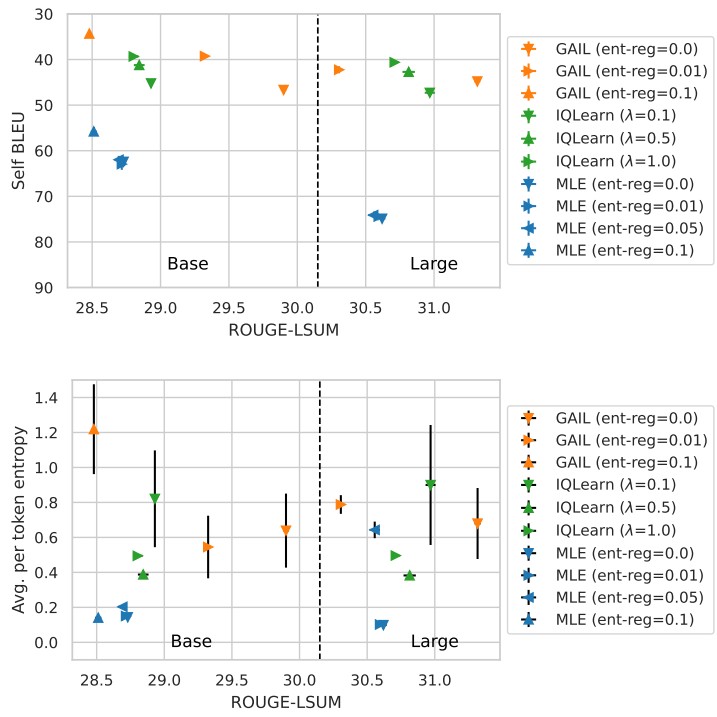

Figure 9: XSUM results for models trained with MLE, IQLearn, and GAIL across different regularization strengths. Top: we show ROUGE-LSUM performance metric on the x-axes and Self-BLEU diversity measure on the y-axis. Bottom: with the per-token entropy as diversity metric. Error bars indicate the standard error of the mean after repeating the experiment with 3 different seeds. Compare to Figure 3 for more results.

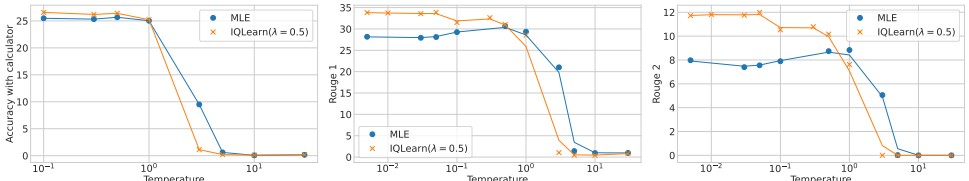

Figure 10: Additional scores against temperature for PALM2 models. From left to right: GSM8k, TDLR (ROUGE-1) and TLDR (ROUGE-2).

black state and has to reach the green on the right. In each of the bottom states the agent has 2 actions, move left or move up. The transition dynamics are noisy so that the agent executes the unwanted action 10% of the time. There are two variants of the MDP, one where the agent can return from the top states by learning to execute the right action and one where it cannot. The latter represents one aspect of concatenation dynamics, the agent cannot return to the exact same state after sampling the wrong action. In a way, it cannot correct its behavior exactly.

When training offline and online variants IQLearn in this setting with demonstrations without mistakes (and their correction), we clearly see that the online version of IQLearn outperforms offline learning by over 11% in success rates, while in the setting without recovery, the difference is considerably smaller and results are within each others confidence bounds with more variance for the offline agent.

### A.3.4 Analyzing GAIL Stabilization

We empirically find GAIL overall more complex to tune and stabilize and aim to provide further insights here. Intuitively, control for language modelling differs from many classical applications via large discrete action space and terminations not being state but action conditioned. In other words, the agent can directly chose to terminate. Therefore the agent can learn to directly terminate if

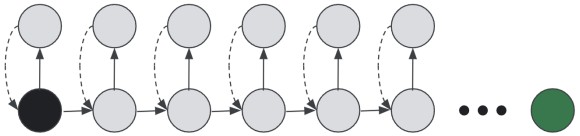

Figure 11: Simplified MDP to represent characteristics of concatenation-based autoregressive generation in comparison to many classical control domains. Dashed lines visualize the potential to return or correct mistakes, missing from autoregressive generation. The agent starts in the black state and has to complete the sequence to reach green to receive rewards.

the discriminator provides negative rewards. If we provide positive rewards, tuning those rewards becomes a complex task as seen in Figure 12, where without additional MLE objective the GAIL agent often uses the maximum sample length with correlated loss of performance (visualized by the drop on ROUGE-1 values). Additional MLE training can help to both shape policy behavior but further also provide more relevant data for the discriminator [55]. Especially in high dimensional action spaces it otherwise becomes challenging to obtain a useful reward signal from the discriminator as seen in prior work [65].

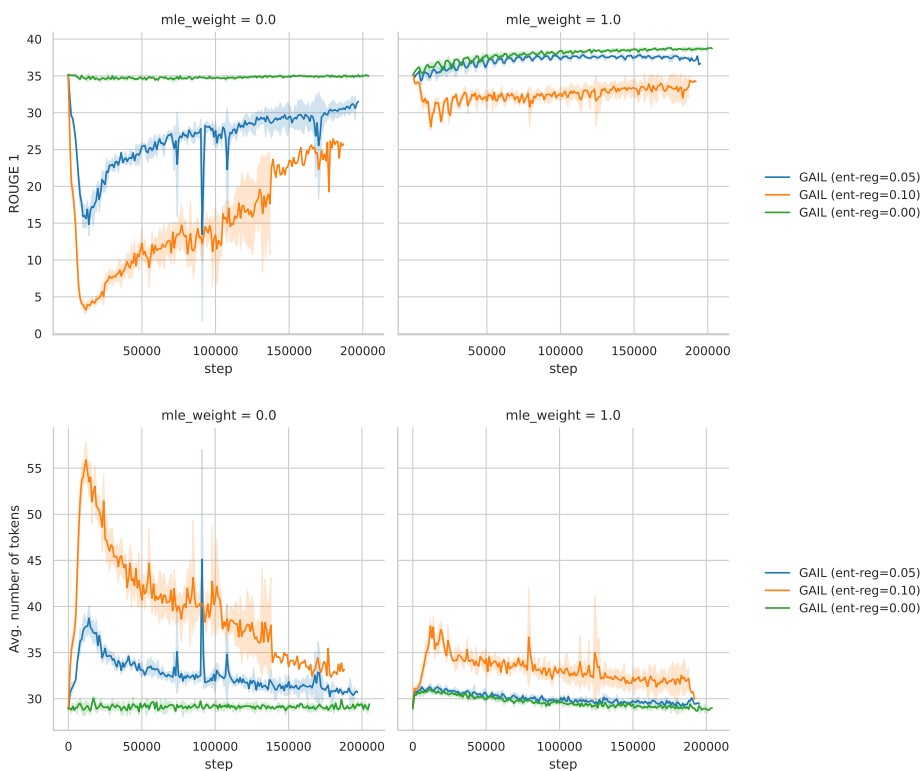

Figure 12: Effect of adding a standard MLE loss (mle_weight=1) on the training data combined with GAIL on XSUM. We show the ROUGE 1 metric and the average length of the generated summaries when training a T5-Large model with GAIL.

## A.4 Expanded Experiments on WMT22

For WMT22, we additionally evaluate IQLearn performance for a wide range of $\lambda$ values. Here, IQLearn consistently gains (up to +2.16 BLEU) over BC until very high values of $\lambda$ (Table 5). Note, that unlike the sampling performance curves in Figure 4, here beam search decoding (size 4) is used with a length penalty 0.6.

Table 5: WMT22 results for offline IQLearn initialised with a PaLM2 checkpoint. *Italic* – best dev BLEU in group (i.e. same mix value), **bold** – best overall.

| $\lambda$ | mixin | dev-BLEU | test-BLEU |
|------|------|----------|-----------|
| 0.0  | 0.0  | 26.92    | 32.34     |
| 0.05 | 0.0  | 29.14    | 34.84     |
| 0.1  | 0.0  | 28.93    | 34.51     |
| 0.3  | 0.0  | 29.07    | 34.79     |
| 0.5  | 0.0  | *29.20*  | 34.63     |
| 0.7  | 0.0  | 28.89    | 35.68     |
| 0.9  | 0.0  | 28.89    | 34.35     |
| 1.0  | 0.0  | 28.92    | 33.69     |
| 0.0  | 0.1  | 27.43    | 32.77     |
| 0.05 | 0.1  | *29.43*  | 34.50     |
| 0.1  | 0.1  | 29.23    | 34.56     |
| 0.3  | 0.1  | 28.89    | 34.84     |
| 0.5  | 0.1  | 28.95    | 34.39     |
| 0.7  | 0.1  | 28.78    | 34.17     |
| 0.9  | 0.1  | 28.83    | 34.41     |
| 1.0  | 0.1  | 28.73    | 34.29     |
| 0.0  | 0.2  | 27.10    | 32.31     |
| 0.05 | 0.2  | 29.08    | 34.40     |
| 0.1  | 0.2  | ***29.46*** | 34.54  |
| 0.9  | 0.2  | 28.85    | 33.99     |

