# OpenReview forum: "Imitating Language via Scalable Inverse Reinforcement Learning"
_NeurIPS.cc/2024/Conference — NeurIPS 2024 poster_

### Official Review · Reviewer_pjSm · 2024-06-19

**Soundness:** 3
**Presentation:** 3
**Contribution:** 2
**Rating:** 6
**Confidence:** 4

**Summary:**

This paper investigates the possibility of applying inverse reinforcement learning for language imitation learning problem. Specifically, the paper reformulates IQLearn as a temporal difference regularized extension of MLE. This basically bridges inverse reinforcement learning with MLE with a coefficient $\lambda$ that can be tuned to change the weight of inverse RL loss versus MLE loss. In experiments on XSUM and GMS8K, the proposed method outperforms vanilla MLE consistently across various base models from 250M to 3B parameters. Time profiling is also reported and it is shown that the proposed offline IQLearn induces minimal computational overhead compared to vanilla MLE.

**Strengths:**

The paper is well-written and easy to follow. The problem is well-motivated as inverse RL is already shown to work better than vanilla MLE is many robotics control problems.

The idea of applying inverse RL to the language setting seems novel to me. However, I am not an expert in inverse RL so not sure about the novelty of reformulating IQLearn as a temporal difference regularized extension of MLE.

The proposed method is extensively validated with various base models on two common language benchmarks: XSUM and GMS8K. The section of time profiling is particularly helpful and it is nice to see offline IQLearn enjoys a similar training efficiency as MLE.

**Weaknesses:**

As admitted in the paper in Line 193 as well, the performance gain is not big (although consistent across different base models) and around 1% accuracy on GMS8K and 1 ROUGE-1 score. It would also be nice to see how this method works on other metrics for text generation such as BertScore[1].

Would be nice to have more intuitive understanding of why adversarial methods such as GAIL do not work very well.

There are some places where the writing can be improved. Please see questions.

[1]Tianyi Zhang, Varsha Kishore, Felix Wu, Kilian Q. Weinberger, & Yoav Artzi (2019). BERTScore: Evaluating Text Generation with BERT. CoRR, abs/1904.09675.

**Questions:**

Line 31, "Including" -> "This includes"

The drawing of Figure 1 can be improved, currently it is hard to tell from the figure what the differences between MLE, offline IRL, and online IRL are. It seems that MLE and offline IRL are exactly the same while online IRL has some blue boxes instead of grey boxes. It is unclear to me what that means even with the help of the caption.

It seems that adversarial imitation learning method GAIL is always outperformed by non-adversarial one IQLearn? What might be a reason for that?

In Figure 2 and 3, is the result of IQLearn offline or online?

**Limitations:**

Limitations are discussed in the Discussion section.

---

> ### Author Rebuttal · Authors · 2024-08-06
>
> Thank you for the constructive feedback. By addressing these helpful comments and questions, the paper has improved and gained in clarity. Please mention further questions or clarifications and we will work to address them ASAP.
>
> As the review mentions, our small performance gains are consistent and considerably larger in some domains, in particular for GSM8k. They further go hand-in-hand with crucial diversity improvements across tasks. We took this feedback seriously and additionally looked into further analysis as well as additional tasks. In addition to XSUM, GSM8k, and TLDR, we added WMT22 results (with results in the general rebuttal). We further analyzed the extracted rewards as described in the general rebuttal and are able to demonstrate correlation between learned rewards and task performance metrics. The particular suggestion of BertScore as an additional metric is highly appreciated. We are looking into experiments with BertScore but will unlikely be able to generate results before the rebuttal session ends due to the computational cost of recreating experiments.
>
> Adversarial imitation is generally known to be less stable (e.g. [1,2]) and often additional heuristics are required for stabilization [3] while hyperparameter tuning becomes increasingly expensive. Due to these aspects, we have only been able to obtain results for the T5 models. This is mentioned in parts of the paper already (see e.g. Sections 3.1 and A.3.3) and we have further expanded the corresponding sections in the submission.
>
> Minor points:
> - We have further expanded our discussion of related work to include the suggested papers.
> - We have improved figure 1 and the corresponding caption.
> - We have fixed the remaining spelling and other mistakes. Thanks for pointing these out.
> - The main IQLearn experiments are offline as we mention in sections 2 and 3. We have emphasized and clarified this throughout all sections.
>
> [1] On the Algorithmic Stability of Adversarial Training, NeurIPS 2021, Yue Xing, Qifan Song, Guang Cheng
>
> [2] Language GANs falling short, ICLR 2020, Massimo Caccia, Lucas Caccia, William Fedus, Hugo Larochelle, Joelle Pineau, Laurent Charlin
>
> [3] Creating Multimodal Interactive Agents with Imitation and Self-Supervised Learning, 2022, DeepMind Interactive Agents Team: Josh Abramson, Arun Ahuja, Arthur Brussee, Federico Carnevale, Mary Cassin, Felix Fischer, Petko Georgiev, Alex Goldin, Mansi Gupta, Tim Harley, Felix Hill, Peter C Humphreys, Alden Hung, Jessica Landon, Timothy Lillicrap, Hamza Merzic, Alistair Muldal, Adam Santoro, Guy Scully, Tamara von Glehn, Greg Wayne, Nathaniel Wong, Chen Yan, Rui Zhu

---

> > ### Comment · Reviewer_pjSm · 2024-08-08
> >
> > Thank the author for the rebuttal. Some of the concerns are clarified. However, the improvements in the main result are still not compelling enough with around 1% across many benchmarks at the cost of a more complicated method and more hyperparameters. I will therefore stay with my original score.

---

> > > ### Author Response · Authors · 2024-08-10
> > > **Additional Information for Official Comment by Reviewer pjSm**
> > >
> > > Thank you very much for the quick response. We appreciate the ability to continue this discussion. Trading off complexity and performance improvements is highly important and we would like to contextualize our contributions with this background.
> > >
> > > 1) The IQLearn offline variant used across most of our experiments does not require online sampling and has a single extra hyperparameter which can be set to 0.1 for most tasks with reasonable gains.
> > >
> > > 2) A key advantage of IRL in comparison to MLE based training is the extraction of reward information which can provide various opportunities for future research and in particular increasing the robustness of RLHF via better reward models.
> > >
> > > 3) The single digit absolut performance gains should be seen in relation to SFT performance. Relative gains are more crucial, e.g. between 10-20% for the PaLM2 models and up to 30% for the smallest T5 models for GSM8k. Furthermore, please be aware that the initial pretrained models already have much greater than 0 performance and are shared across methods such that we only change the algorithm for a comparably small amount of overall training data (due to the computational cost of pretraining experiments).
> > >
> > > 4) It is correct that some domains have smaller improvement, in particular XSUM. Including them serves 2 purposes. They demonstrate impact on increased diversity but also enable us to discuss IRL benefits in relation to task/data properties; it is more suited for some tasks than others.
> > >
> > > 5) In general, we add strong diversity improvements in addition to performance which can be highly relevant for creative applications as well as for online data generation such as in RLHF.
> > >
> > > Given the simplicity in the offline setting (Pt 1) with a single hyperparameter, these aspects represent considerable advantages.

---

### Official Review · Reviewer_UqkB · 2024-07-11

**Soundness:** 4
**Presentation:** 3
**Contribution:** 4
**Rating:** 8
**Confidence:** 3

**Summary:**

The paper introduces a RL-centric perspective to imitation for LLMs, with an novel imitation learning algorithm for fine tuning LLMs, that is derived from IQLearn, forming a dynamics dependent temporal difference regularized variant of MLE. The authors provide an extensive analysis of other potential inverse RL algorithms for imitation learning with LLMs, and demonstrate IQLearns ability to perform well at tasks whilst still having good diversity. Their approach is a promising offline IRL approach.

**Strengths:**

* Clarity: The introduction is clear and well motivates the problem. Overall the paper has good clarity, bar a few typo's.
* Originality: The RL-centric perspective to imitation for LLMs and the IQLearn objective and formulation appears novel and significant to the community.

**Weaknesses:**

* The paper could benefit from more experimental tasks such as CommonGEN, ROCStories, EMNLP2017, and COCO, as done in TextGAIL.
* Missing error bars in Figure 4 and Figures 2 & 3 could benefit from error bars as well, perhaps with a different version in the appendix.
* Minor: The first paragraph could benefit from references to ground the reader.
* Minor: L90. Reading the references cited in this paragraph, they do not mention the term $(1-\gamma)$; perhaps find a reference for it, remove this term, or explain its inclusion in a footnote or an appendix.
* Minor: Equation 8 could put an enclosing bracket for the min operator to help guide the reader.
* Minor: The presentation of the figures could be improved by making the font size larger and using vector graphics.

Typos:

* L40: “behavior cloning [7], its” -> “behavior cloning [7], and its”
* L123: “further simplify” -> “further simplify it”
* L168: “temporal different regularization” -> “temporal difference regularization”
* L174: “we a short” -> “we use a short”
* L174: “pure MLE beneficial” -> “pure MLE being beneficial”
* L:202: “full trajectories” -> “full of trajectories”
* L:218: “Right” -> “Left”

**Questions:**

* Are the T5 model’s pre-trained? I presume they are, if so it could be helpful to state in the main paper in section 3.2.1 that the T5 models are pre-trained, and details of this.
* As stated, IQLearn with Lambda = 0 “retrieves standard MLE.” Can we then interpret Figure 2 as IQLearn (Lambda=0.0) as MLE? If so, it is interesting how IQLearn always outperforms MLE, even for small Lambda. Could you perform ablations for small non-zero values of Lambda to show if it converges with MLE on the task results?
* How do the conclusions of Figure 4 hold if Lambda is varied? What is the expectation?
* Figure 6: can you quantify the uncertainty estimate used?
* How would you see future work extending your approach to use human pairwise preference datasets? That is combine approaches perhaps such as Direct Preference Optimization.

**Limitations:**

These were adequately discussed in Section 5.

---

> ### Author Rebuttal · Authors · 2024-08-06
>
> Thank you for the constructive feedback. By addressing these helpful comments and questions, the paper has improved and gained in clarity. Please mention further questions or clarifications and we will work to address them ASAP.
>
> Uncertainty estimates and error bars: We always plot the standard error (standard-deviation / sqrt(number-of-seeds) in all plots. We added a comment to the paper explaining our uncertainty estimation. Figures 2 & 3 already include error bars; these are very small due to only minimal variability from the random seeds. We updated the color of these error bars to make them more legible. Due to computational requirements, we cannot add further experiments to add error bars for Figure 4 before the rebuttal deadline, but have added it to our prioritization list to follow-up when other experiments are completed.
>
> We strongly acknowledge the benefits of extending analysis and a broader task set. While experiments, in particular with hyperparameter sweeps and multiple seeds are expensive, we have added results on WMT22 training (with results in the general rebuttal). We continue looking into further experiments that can be performed with limited costs and have, in parallel, extended our analysis to obtain further insights without additional training runs. For this, we have investigated the rewards extracted by IRL as described in the general rebuttal comment.
>
> Regarding the extension to preference learning, this is a very good point and we quickly discuss opportunities in the discussions section. A key benefit is the ability to extract reward information from both demonstration and preference datasets, which could bring benefits regarding the robustness of RLHF. While these are untested for language modeling, there exist early examples for classic robotics tasks [1].
>
> Ablations on \lambda to interpolate between MLE and IQLearn are of crucial interest and included (with limited settings), for example in Figure 2. We know from early tests that much smaller values critically reduce performance and \lambda = 0 algorithmically reduces to pure MLE. However, since these experiments are computationally expensive we are unable at this point to add a more detailed study.
>
> Similarly, we made the decision to sweep over the quality-diversity tradeoff for PaLM2 models via inference time settings, rather than separate training experiments, to minimize computational costs and provide better coverage of different mechanisms for the tradeoff. We agree this would be interesting to investigate, but have to rely on the related sweeps for Figures 2 and 3 for the time being.
>
> Minor points:
> - The T5 models are in fact pretrained. Thank you for pointing this out; we have expanded the model discussion in the paper and clarified this aspect.
> - We have fixed the remaining spelling and other mistakes, as well as added further clarifications as suggested. Thank you for pointing these out.
>
> [1] Learning Reward Functions from Diverse Sources of Human Feedback: Optimally Integrating Demonstrations and Preferences, 2020, Erdem Bıyık, Dylan P. Losey, Malayandi Palan, Nicholas C. Landolfi, Gleb Shevchuk, Dorsa Sadigh

---

> > ### Comment · Reviewer_UqkB · 2024-08-10
> >
> > I thank the authors for their detailed rebuttal response, clarifications and improvements to the paper. The new results on WMT22 are a welcomed addition. As my concerns have been addressed I have raised my score.

---

### Official Review · Reviewer_6J5H · 2024-07-11

**Soundness:** 3
**Presentation:** 2
**Contribution:** 3
**Rating:** 5
**Confidence:** 3

**Summary:**

This paper investigates using inverse reinforcement learning (IRL) to directly optimize sequences for fine-tuning large language models. Moreover, this work reformulates inverse soft-Q-learning as a temporal difference regularized extension of maximum likelihood estimation (MLE), which bridges IRL and MLE in supervised fine-tuning tasks. The experiments demonstrate the clear benefits of IRL-based imitation for retaining diverse responses while maximizing task performance.

**Strengths:**

1.	This work investigates the potential of inverse reinforcement learning for tuning LLMs. The strengths of IRL are evaluated in terms of performance, diversity, and computational requirements. The experiments demonstrate the computationally cheaper offline IRL can obtain crucial performance gain over MLE-based LLM tunning.

2.	IQLearn is a method that can work online and offline. The re-formulation of IQLearn to LLM tunning enables large performance gains compared to MLE methods. Moreover, IQLearn can be regarded as a regularized extension of MLE, which bridges IRL and MLE for LLM tunning.

**Weaknesses:**

1.	IRL methods, even for offline IRL methods, cost more time than MLE for LLM tunning.

2.	More experimental details are required to reproduce the results.

**Questions:**

1.	The original GAIL is an online IRL method, does GAIL execute offline or online in the experiments?

**Limitations:**

The discussion section discussed the potential limitations of this work.

---

> ### Author Rebuttal · Authors · 2024-08-06
>
> Thank you for the constructive feedback. By addressing these helpful comments and questions, the paper has improved and gained in clarity. Please mention further questions or clarifications and we will work to address them ASAP.
>
> Thank you for pointing out computational requirements. We crucially reduce computational costs by training offline and thus removing the additional sampling cost for IQLearn in all sections but the online to offline comparison in section 3.3.1. The remaining difference in computational costs are partially due to a non-optimized implementation in our codebase, which shares many components between offline and online IRL algorithms. We have further emphasized the implementation dependence in the paper. We have started to recreate a pure and clean offline implementation but will be unable to verify results with the new implementation before the rebuttal ends.
>
> We have added more general details on the experiments. We would be glad to provide further clarifications and experiment details if the reviewer could specify what additional information or results would be needed from their perspective.
>
> We have further expanded the results, adding WMT22 where offline IQ-Learn shows increased performance in comparison to MLE-based training (with results in the general rebuttal). Finally, a key benefit for IRL-based models lies in additionally extracting reward information. To explore this, we have added analysis for the correlation between extracted rewards and performance metrics, with results summarized in the general rebuttal.
>
> Minor points:
> - GAIL is an online method that requires training the discriminator between the online agent and offline demonstration dataset. We have emphasized this in the paper.

---

> > ### Comment · Reviewer_6J5H · 2024-08-12
> >
> > Thank the authors for the rebuttal. I maintain my score as is.

---

> > > ### Author Response · Authors · 2024-08-12
> > > **Additional Information for Official Comment by Reviewer 6J5H**
> > >
> > > Thank you for the quick response. Our team has worked hard to answer your questions and address your comments. Could you please provide further information or open questions underlying the kept rather than updated score to enable us to address them.

---

### Official Review · Reviewer_aRfw · 2024-07-11

**Soundness:** 2
**Presentation:** 2
**Contribution:** 2
**Rating:** 4
**Confidence:** 3

**Summary:**

In this paper the authors aim to cast fine-tuning LM as inverse RL from the perspective of distribution matching. In order to avoid online generation in existing IRL algorithms such as GAIL, the authors leverage an offline IRL algorithm, IQL, and reformulate it as a temporal difference regularized extension of MLE. This derivation enables the use of expert samples solely in the form of distribution matching. Experiments in XSUM and GSM8K show that their method outperforms both MLE and online IRL algorithms in terms of task performance and diversity.

**Strengths:**

The reformulation of IQL into distribution matching is novel and it leads to a simple and efficient version of IQL. The simplicity of the new derivation of IQL may potentially make it an impactful learning objective like MLE.

**Weaknesses:**

- The improvement is somewhat marginal especially in the task performance on the summarization benchmark.
- It is not clear the proposed method is better than MLE in terms of quality-diversity trade-off.
- The diversity evaluation for lower open-endedness tasks like GSM8K does not make much sense.

**Questions:**

- Could the authors provide an analysis for quality-diversity trade-off [1] of the algorithms?
- Why does the temporal regularization term encourages higher diversity?
- It would be helpful if the paper contained pseudo code of the proposed algorithm.

[1] LANGUAGE GANS FALLING SHORT

**Limitations:**

see weaknesses

---

> ### Author Rebuttal · Authors · 2024-08-06
>
> Thank you for the constructive feedback. By addressing these helpful comments and questions, the paper has improved and gained in clarity. Please mention further questions or clarifications and we will work to address them ASAP.
>
> One of the review’s key points regards limited performance improvements on summarization. Here, we would like to emphasize that the goal of our paper is to establish in which tasks IRL-like learning is most impactful. Overall, the performance gains are stronger across the other domains, and in summarization we additionally see considerably improved diversity of generations. We have further emphasized the limitations of MLE/BC with respect to long target/trajectory lengths known from the imitation learning literature (e.g. DAgger [A]).
>
> To further improve the empirical evaluation, we have added results on WMT22 where offline IQ-Learn shows increased performance in comparison to MLE-based training (with results in the general rebuttal). Finally, a key benefit for IRL-based models lies in additionally extracting reward information. To explore this, we have added analysis for the correlation between extracted rewards and performance metrics, with results summarized in the general rebuttal.
>
> Regarding the connection between IRL and higher diversity, equation 3 in the paper describes this best. The original state-distribution-matching formulation builds on entropy-regularized RL and in particular includes a term for causal entropy. We have further emphasized this connection in the paper.
>
> Thank you in particular for enabling us to strengthen the connection to [1]. Our temperature sweeps as well as the sweep over different entropy regularisations enable the generation of the kind of quality diversity evaluations that [1] is arguing for. While we move away from purely adversarial inverse RL and expand to saddle-point-based methods like IQLearn, their insights are highly relevant and we have clarified additional heuristics required to enable better performance for our implementation of GAIL (which are already included in the appendix but were previously skipped in the main paper).
>
> Minor:
> - We add pseudo code to the appendix to minimize required space in the main paper.
> - We further expanded our discussion of related work to include the suggested paper.
>
> [A] A Reduction of Imitation Learning and Structured Prediction to No-Regret Online Learning, AISTATS 2011, Stéphane Ross, Geoffrey J. Gordon, J. Andrew Bagnell

---

> > ### Author Response · Authors · 2024-08-13
> > **Follow up for reviewer aRfw**
> >
> > Thank you for your original review. Our team has worked hard to answer your questions and address your comments. Since the original rebuttal, we further expanded the reward analysis to GSM8k and TLDR.
> >
> > With the discussion ending soon please ask any remaining questions or comments soon and we will answer ASAP; or share if previous answers and additional experiments have addressed your concerns.

---

### Official Review · Reviewer_tAhh · 2024-07-15

**Soundness:** 2
**Presentation:** 2
**Contribution:** 2
**Rating:** 4
**Confidence:** 3

**Summary:**

The paper looks at the existing IQLearn algorithm and makes some connections with standard maximum likelihood learning in the context of sequence based language models.

On the empirical front, results are given showing different model performance along diversity (measured via self-Bleu) and some accuracy measure (such as rouge). These are given on a few public datasets. The empirical part of the paper lacks focus on what is trying to be optimised, and the actual metrics that are reported leave open some potential that the performance is not necessarily better. For example self-bleu may increase, but it isn't clear this is a better model. The diversity may be erroneous, but still more diverse and still scoring well on the token overlap measures of rouge etc.
The other graphs and figures fail to leave the reader with a clear story of why using imitation learning at the sequence level, rather than MLE or RLHF (imitation learning at the token level) is necessarily better.

**Strengths:**

* A solid theoretical analysis is given which draws links between MLE and forms of IRL.

**Weaknesses:**

* The story of the paper is not that clear, and the empirical results do not tell a clear story. The paper shows that the IRL methods can be used to obtain better models on the Pareto type plots of quality and diversity, however it's not clear that these are actually better models.

* The data used in the empirical section only appears a long way into section 3. It would be much clearer if the empirical section had a clearer introduction of what types of problems are being looked at, and what data is being used for such.

**Questions:**

What was the conclusion for which tasks and dataset properties is IRL most relevant for? (This is a question at line 160, however it's not clear that it is answered).

**Limitations:**

No concerns.

---

> ### Author Rebuttal · Authors · 2024-08-06
>
> Thank you for the constructive feedback. By addressing these helpful comments and questions, the paper has improved and gained in clarity. Please mention further questions or clarifications and we will work to address them ASAP.
>
> The included experiments show that IRL methods (GAIL/IQLearn) improve both performance and, crucially, diversity metrics, compared to common next token likelihood. The performance improvements are small for XSUM but consistent across domains. If the reviewer could specify what additional information or results would be needed to fulfill specific additional criteria for 'actually better models', we would be glad to provide further clarification. In addition, in the paper we have clarified the exact metrics used, and how to interpret the results. In particular for Self-BLEU, we will expand the captions to emphasize that smaller numbers indicate better diversity (lower self-similarity) and that the y-axis is flipped.
>
> Furthermore, if the point above is meant in relation to the comment “The empirical part of the paper lacks focus on what is trying to be optimised”, we can clarify. Section 2 describes the optimization (aspects such as the relation of reverse KL to MLE, regularized chi2 for Iq-learn). However, evaluating these divergences once trained is hard, because we do not have access to the distribution underlying the dataset. In classic imitation learning papers, the underlying ground truth reward function can be used for evaluating the quality of the imitating agent (typically, the expert is an RL agent trained on some known reward). Yet, in our case, we do not have such a reward function, and the expert is a human. We therefore have to rely on these proxy metrics, and we consider classic ones in the context of language models.
>
> In parallel to clarifying this point, we have expanded our experiments to include results on WMT22 (with results in the general rebuttal). Furthermore, IRL-based models have the added benefit of extracting reward information from demonstration data. We have added further analysis on the correlation between extracted rewards and performance metrics to visualize this advantage, with exact numbers added in the general rebuttal.
>
> Regarding the question of most relevant tasks for IRL, we have emphasized the theoretic connection to target/trajectory length known through the imitation learning literature (e.g. DAgger[1]). Practically, the performance gains are stronger with longer target lengths (GSM8k) and partially with smaller datasets (ablations on smaller subsets of XSUM), while improved diversity of generations is persistent across tasks. We have expanded our discussion of these aspects in the paper.
>
> Minor points:
> - We will introduce the used datasets and tasks earlier in Section 3.
> - RLHF: We do not provide an alternative to RLHF but rather emphasize the RL perspective during SFT, which could in the future enable better connections between SFT and RLHF data sources. As both method types use different data sources, a direct, fair comparison between IRL-based approaches and RLHF is not possible. Instead, these are complementary.
>
> [1] A Reduction of Imitation Learning and Structured Prediction to No-Regret Online Learning, AISTATS 2011, Stéphane Ross, Geoffrey J. Gordon, J. Andrew Bagnell

---

> > ### Author Response · Authors · 2024-08-13
> > **Follow up to reviewer tAhh**
> >
> > Thank you for your original review. Our team has worked hard to answer your questions and address your comments. Since the original rebuttal, we further expanded the reward analysis to GSM8k and TLDR.
> >
> > With the discussion ending soon please ask any remaining questions or comments soon and we will answer ASAP; or share if previous answers and additional experiments have addressed your concerns.

---

### Author Rebuttal · Authors · 2024-08-06

We would like to thank our reviewers for their valuable feedback. The paper has already considerably improved and we will work hard to address further comments and questions to conclude the rebuttal process to everyone’s satisfaction.

The reviews generally appreciate the discussion of inverse RL in the context of language modeling as well as the derivations and reformulation of IQLearn as temporal difference regularized MLE. Multiple reviews ask for the addition of further tasks and, while the computational cost is non-negligible, we have added results on the large WMT22 English-German task (285M examples). IQLearn gains are between 2.2-2.4 over MLE with beam search decoding. Early results with computationally cheaper decoding via temperature sampling indicate larger gains (evaluation is ongoing and we will add the complete results via comments ASAP). We are investigating further options for tasks as well.

We aim to address questions around the key benefits of IRL. Here, in addition to training a policy (i.e. the generative LLM), IRL enables the extraction of reward functions from demonstration data. We discuss multiple benefits for this additional source of reward information in the discussion section and this includes the potential to mitigate reward model overoptimization during RLHF [1], and we have added more analysis on this (see below). In IQLearn, in particular, we can recover rewards from the Q-function, as we describe in line 136 (via $r_t=Q(s_t, a_t) - \gamma V(s_{t+1})$). We have added correlation analysis between the extracted rewards and task performance metrics. As an example, current results show both Pearson Correlation Coefficient and Spearman's Rank Correlation Coefficient between 0.21 and 0.44 for IRL-extracted rewards and BLEU and between 0.27 and 0.48 for ChrF on WMT. For comparison, the corresponding correlations for MLE are respectively around 0.04-0.05 and 0.004-0.005. The complete results have been added to the paper. PDF updates are unfortunately not possible during the rebuttal process, so please reach out if there are further questions about these results.


Further specific details for each review are included in our individual answers. Please do not hesitate to point out further questions or comments.

[1] Scaling Laws for Reward Model Overoptimization, 2022, Leo Gao, John Schulman, Jacob Hilton

---

### Decision · Program_Chairs · 2024-09-25

**Decision:**

Accept (poster)

**Comment:**

This paper proposes a technique based on Inverse Reinforcement Learning for LLM training, as alternative to the standard MLE estimator for next token prediction. The proposed approach, that builds on existing work on inverse soft-Q learning, is shown to help increase response diversity while maintaining or improving quality (depending on the benchmark).

Reviews were somewhat mixed (4,4,5,6,8), with reviewers being generally appreciative of the novelty of the approach (applying IRL to LLM training), but with an overall concern towards empirical results that do not show strong improvements over standard MLE across all benchmarks. In their response, the authors added more experiments showing benefits on the WMT22 task.

Among the two reviewers who rated 4, one didn't follow-up on the author response, while the other did (privately) and decided to keep their score due to quantiative improvements not being strong enough.

Although I agree it would be nice to see stronger improvements across the board, I consider the gains in diversity to be significant and potentially quite relevant in some applications. In addition, being able to improve on some tasks (while not degrading on others) remains meaningful, and I believe this work also has potential to encourage further research on IRL for LLM training. As a result, I am recommending acceptance.